# Optoelectronic analysis of technical factors and performance of elite-level air pistol shooting

Daniel Mon-López[1]*, Alfonso de la Rubia[1], Abraham García-Aliaga[1], Jorge Acebes-Sánchez[2], Ignacio Refoyo Roman[1], Jorge Lorenzo Calvo[1]

1 Facultad de Ciencias de la Actividad Física y del Deporte (INEF—Sports Department), Universidad Politécnica de Madrid, Madrid, Spain, 2 Faculty of Health Sciences, Universidad Francisco de Vitoria (UFV), Pozuelo de Alarcón, Spain

* daniel.mon@upm.es

**Data Availability Statement:** All relevant data are within the paper and its Supporting Information files.

## Abstract

Technical elements are related to shooting performance; however, the importance of each factor regarding performance, especially in elite-level pistol shooters, remains controversial. The objective of this study was to determine the technical factors that influence pistol performance. One elite-level shooter was analysed during the season before the Olympic classification European championship through a total of six competitions (n = 360 shots). Aiming point trajectories were measured with the SCATT optoelectronic system. Variables were classified into six categories: performance; aiming time; stability of hold; aiming accuracy; cleanness of triggering and shooting delta. Principal component analysis, multiple regressions, Pearson correlations and ANOVAs were used to analyse the data. The results showed that five components (aiming time, stability of hold, aiming accuracy, cleanness of triggering and shooting delta) determined a total of 79.68% of the shooting variance. Specifically, aiming accuracy and cleanness of triggering explained up to 25% of the shooting score, with cleanness of triggering serving as the determining factor. Correlations were found among the performance and stability of hold, aiming accuracy, cleanness of triggering and shooting delta. Last, significant differences were found among 8-score, 9-score and 10-score shots. We conclude that while aiming accuracy seems to be related to the performance, cleanness of triggering could be the most critical technical element; furthermore, while stability of hold does not seem to be a determining factor of the score, it could be a general prerequisite to achieve high levels of performance in elite-level air pistol shooters.

## Introduction

Shooting is a multifactorial process that occurs simultaneously, in which performance seems to be the sum of independent factors [1]. Specifically, in air pistol shooting, many factors have been described as performance predictors. Thus, physical factors, such as balance or strength [2], and psychological variables, such as anxiety [3] or self-control [4], could be related to

**Funding:** The author(s) received no specific funding for this work.

**Competing interests:** The authors have declared that no competing interests exist.

shooting performance. Nonetheless, previous studies have indicated that the most important performance factor are the technical elements [1].

Previous scientific literature has carried out principal component analysis to determine the main components of shooting variance. Reports indicated that six components (aiming time, stability of hold, measurement time, cleanness of triggering, aiming accuracy, and timing of triggering) explained 88% of the total variance in air rifle shooting [5]. Moreover, Hawkins [6] pointed out that 3 components specific to air pistol (hold stability, time and aim accuracy) accounted for a total variance of 81.07%, while Olsson and Laaksonen [7] identified five key components (aiming time; stability of hold; aiming accuracy; cleanness of triggering and timing of triggering) that accounted for a total variance of 75.8%. Surprisingly, the percentage of variance is similar in the literature; however, a consensus has not been reached on the main components.

One of the main shooting factors is the stability of hold or weapon stability, considered as the steadiness of the pistol barrel and measured as the mean aiming trace speed or the percentage of aiming time spent inside the 10-ring or 9-ring drawn around the hit point during the last second [5, 8]. This stability of hold could be a differential element of the sport level [5] and is related to body sway in shooting, although this correlation seems to depend on the modality [9] and on each individual shooter [10]. Moreover, elite-level shooters have better stability than novice shooters for air rifle [5] and air pistol [11]. However, although the correlation between postural balance and hold stability is strong, the direct effect on performance may be less than 1% of the performance variance in the air rifle [5], or it may not have an interindividual effect in air pistol [10]. In contrast, body sway had a direct effect on performance in junior air pistol shooters and less-experienced shooters [12].

Regarding the stability of the hold, this factor could explain up to 54% of the score variance in air rifle shooting [5]. However, stability of hold could explain 33% of the score variance in male pistol shooters [6, 13], while in a mixed gender sample, this variance would be between 24% and 36% depending on the selected variable [7]. Additionally, as the aim-point fluctuation increased, the errors also increased and the score decreased, with significant correlations observed for 3/5's of the elite-level pistol shooters [10]. In contrast, the stability of the hold had small effects on performance in running target shooting [14].

Another important variable is the aiming accuracy. It has been suggested that the percentage of aiming time spent inside the 10-ring during the last second is associated with performance in elite-level pistol shooters [10]. Thus, aiming accuracy variables would be related to pistol performance ($r$ = .32 to .42) [6]. However, recent studies have found that aiming accuracy is a more important determinant for air pistol ($r$ = .65 to .74) [7] and air rifle performance ($r$ = .64) [15].

Interestingly, there is some controversy regarding the relevance of cleanness of triggering (movement of the aiming point during the last 0.2 s [5] or the distance between the midpoint of aim during the last interval time selected and the centre of the shot [16]) on performance. Thus, cleanness of triggering explained 47% of the variance in the air rifle and 37% in running target shooting [14]. Similarly, strong correlations were found in the air rifle, with higher values for international than national level shooters [5]. Additionally, the changes in cleanness of triggering during three-seasons were related to the performance of elite-level rifle shooters [17]. In contrast, Olsson and Laaksonen (7) showed that cleanness of triggering was negatively related to performance ($r$ = -.48). In this line, weak corelations were found between the NOP-TEL relative triggering value and the pistol score [6, 13].

The last technical aspect is related to the time to shoot, which has two components: timing of triggering (total aiming time) and time on target (aiming time spent continuously on the target). Interestingly, some variables of shooting time were related to air rifle performance

(aiming time spent inside the 7-ring), while others were not (time on target or total time), and no differences were found between shooting levels [5]. In contrast, the total time spent on the target was a discriminating element between sport levels in air rifle [14]. Specifically, in air pistol, the time on target has typically been established as being optimal from 5 to 10 s [18] or 4 to 10 s [8]. In addition, Hawkins (6) did not find any correlations between time variables and shooting performance. Similarly, Olsson and Laaksonen (7) did not find a relation between time on target and performance, but they did observe a relationship between the timing of triggering (NOPTEL) and performance. In contrast, other studies pointed out that the total sighting time was negatively related to absolute shot displacement (sum of vertical and horizontal displacement) [19].

Unfortunately, many questions remain unanswered, and some authors have suggested the need for additional studies with elite-level shooters [1], especially when considering that Olympic shooting is a maximum precision sport where minimal errors can lead to score decreases [20]. In this line, shooting performance is measured in two stages during air pistol competitions: the qualifying and the final rounds. Interestingly, performance is measured differently in these stages. Thus, during the qualifying round, the score is measured by points without decimals, while during the final round, the score is measured by points with decimals [21]. However, most of the previous literature has analysed air pistol performance using only points with decimals [7, 10, 13].

Additionally, optoelectronic devices have been mainly used to measure technical elements during trainings or simulated competitions [1, 7]. NOPTEL [14] and SCATT [8] are the two most commonly used optoelectronic devices. However, although both systems are widely used by coaches, the number of SCATT studies is still very limited compared to that of NOPTEL [1].

In consequence, it seems necessary to determine the performance rates of variance in air pistol with accuracy. Additionally, the number of studies using the SCATT optoelectronic device is very limited, and few performance analyses have been performed using points with and without decimals. Consequently, the main objective of the present study was to analyse what factors determine shooting performance in an elite-level air pistol shooter by using the SCATT optoelectronic device.

## Materials and methods

### Participant

The participant in this study was one elite-level pistol shooter. Two inclusion criteria were used to consider the elite-level level of the shooter: 1) Ranked in the top ten at the European Championship of Wroclaw, 2020. 2) Achieved a result equal to or over 580 points. This score was selected to ensure a world-level score because it was the mean score to be a finalist in the previous year World Cups (New Delhi 580, Beijing 581, Munich 581, and Rio de Janeiro 578) [21]. The study was performed in accordance with the ethical standards of the Helsinki Declaration and the participant signed an informed consent before the data collection. This study was approved by the ethics committee of the Polytechnical University of Madrid.

### Apparatus and variables

Shooting aiming point trajectories were measured with the optoelectronic system SCATT, model MX-W2 (SCATT Electronics LLC, Russia, Moscow). The training device consisted of an optical transmitter-receiver unit, with a weight of 56 g and dimensions of 34x35x60 mm. The software version used was "Scatt expert". The SCATT device was connected to a Microsoft Surface Pro 7 laptop (Microsoft Corporation, Washington, USA). Shooting scores and aiming

point trajectories were recorded for every competition shot at a 50 Hz sampling rate. Data were recorded during the second prior to the shot. The ballistic ratio or $F$ coefficient was adjusted to 12. The ballistic ratio or the $F$ coefficient is primarily a determinant of how much the pellet gets 'thrown', and the muzzle is moving in this direction at the instant of firing the shot before impacting the target. The manufacturers quoted that the accuracy of the SCATT system was ± 0.1 mm.

The registered variables were distributed into six categories (Table 1): overall performance; stability of hold; aiming accuracy; cleanness of triggering; time on target and shooting delta. All variables are described using the mean ± standard deviation (Table 2).

## Methodology

Six training competition tests of 60 shots were recorded with a mean performance of 578,16 ± 3,76 (minimum 573- maximum 583) points without decimals. Competitions were performed using electronic targets under the ISSF official competition rules and regulations [22]. In addition, the shooter was allowed to make unlimited shots during the 15 minutes of preparation time before the competition to adjust the SCATT system and to use his own shooting equipment. Simultaneously, the SCATT system was used to register the shooting variables. Only shooting score and hit placement in the SCATT were displayed to the shooter, like a real-life competition situation. The data collection was carried out during the preparatory period for the European Championship Wroclaw (2020).

**Table 1. Descriptions of the shooting technical variables.**

| Component | SCATT Variable (unit) | Description |
|---|---|---|
| Overall Performance | Shooting score with decimals (pts) | Shot score measured in points with decimals and ranged between 0–10.9 |
| | Shooting score without decimals (pts) | Shot score measured in points without decimals and ranged between 0–10 |
| Stability of hold | STD S1 (mm/s) | Mean aiming trace speed for the control interval (1 s before the shot) |
| | STD S2 (mm/s) | Mean aiming trace speed for the control interval (0.25 s before the shot) |
| | STD 10.5a0 (%) | Percentage of aiming time spent inside the 10.5-ring drawn around the point during the last second |
| | STD 10a0 (%) | Percentage of aiming time spent inside the 10-ring drawn around the point during the last second |
| | STD 9a0 (%) | Percentage of aiming time spent inside the 9-ring drawn around the point during the last second |
| Aiming accuracy | ACC10.5 (%) | Percentage of aiming time spent inside the 10.5-ring during the last second |
| | ACC10.0 (%) | Percentage of aiming time spent inside the 10-ring during the last second |
| Cleanness of triggering | DA (mm) | Distance between the midpoint of aim during the last second and the centre of the shot |
| | DA250 (mm) | Distance between the midpoint of aim during the last 0.25 second and the centre of the shot |
| Time on target | TIME ON TARGET (s) | Aiming time spent continuously on the target |
| Shooting Delta | Δ S1-S2 | Speed difference between STD S1 and STD S2 |
| | Δ HOLD10 (%) | Percentage of difference between AIM ACC 10.0 and STD 10a0 |
| | Δ HOLD10.5 (%) | Percentage of difference between AIM ACC 10.5 and STD 10.5a0 |

**Table 2. Descriptive values of the technical variables by category.**

| Variable category | Variable | M | SD |
|---|---|---|---|
| Performance | Score with decimals (points) | 10.08 | 0.47 |
| | Score without decimals (points) | 9.64 | 0.5 |
| Aiming time | Time on target (s) | 8.6 | 2.43 |
| Stability of hold | STD 9a0 (%) | 99.64 | 1.2 |
| | STD 10a0 (%) | 82.74 | 11.67 |
| | STD 10.5a0 (%) | 35.93 | 13.23 |
| | STD S1 (mm/s) | 108.83 | 19.51 |
| | STD S2 (mm/s) | 108.5 | 31.2 |
| Aiming accuracy | ACC10.0 (%) | 60.22 | 21.72 |
| | ACC10.5 (%) | 19.87 | 13.37 |
| Cleanness of triggering | DA (mm) | 6.62 | 3.43 |
| | DA250 (mm) | 4.88 | 2.64 |
| Shooting Delta | Δ S1-S2 | -0.33 | 25.82 |
| | Δ HOLD10 (%) | 22.52 | 20.95 |
| | Δ HOLD10.5 (%) | 16.06 | 15.75 |

Notes. *M* = Mean, *SD* = Standard deviation.

## Statistical analysis

The data were described with the arithmetic mean (*M*) and standard deviation (*SD*). The normality of the variables was tested with the Kolmogorov–Smirnov and Shapiro–Wilk tests. One-way ANOVA was used to make comparisons among 8-score, 9-score, and 10-score shots. A Games-Howell post hoc test analysis was performed to determine differences between groups. The effect size was calculated using Hedges' *G* index (*d*; ± 95% confidence interval) and interpreted using three cut off points: small (*d* = 0.2), medium (*d* = 0.5) and large (*d* = 0.8) [23].

A principal component analysis with varimax rotation was used to form orthogonal linear combinations from aiming point variables in air pistol following previous studies [5, 6]. The number of components was determined by a minimum eigenvalue of 0.9 and a minimum of 5% variance accounting for a component [7].

Two-tailed Pearson's correlation coefficient analysis was used to examine the relationship between shooting performance and the technical variables. The correlation effect size was assessed using *r* (small *(r = .10)*; medium *(r = .30)* and large *(r = .50)* [6].

Finally, two four-step hierarchical regressions were performed to analyse the relationships between performance and technical variables. Collinearity statistics were undertaken to examine the linear association between the predictive variables in the stepwise multiple regression analysis. IBM SPSS Statistics software (SPSS 25.0. IBM Corp., Armonk, NY, USA) was used for the mathematical calculations. The level of significance was set at $p < .05$.

## Results

Table 3 shows the comparison results of the technical variables values by shot score without decimals. The score without decimals analysis showed differences between groups as follows: STD 9a0 $F_{(2,357)} = 3.86$; $p = .022$; ACC10.0 $F_{(2,357)} = 14.44$; $p < .001$; ACC10.5 $F_{(2,357)} = 11.68$; $p < .001$; STD 10.5a0 $F_{(2,357)} = 4.38$; $p = .013$; DA $F_{(2,357)} = 41.51$; $p < .001$; DA250 $F_{(2,357)} = 23.17$; $p < .001$ and in Δ HOLD10 $F_{(2,357)} = 9.20$; $p < .001$. For the rest of the comparisons, no differences were found ($p > .05$. The Games-Howell post hoc analysis showed the following

**Table 3. Comparison of the technical variables values by shot score without decimals.**

| | 8-score | | 9-score | | 10-score | | Total number of shots | |
|---|---|---|---|---|---|---|---|---|
| | *M* | *SD* | *M* | *SD* | *M* | *SD* | *M* | *SD* |
| Time on target (s) | 9.18 | 2.14 | 8.79 | 2.27 | 8.49 | 2.51 | 8.60 | 2.43 |
| STD 9a0 (%) | 98.00 | 1.83 | **99.67**[A] | 1.21 | **99.65**[A] | 1.17 | 99.64 | 1.20 |
| STD 10a0 (%) | 75.25 | 11.50 | 81.08 | 12.79 | 83.75 | 10.93 | 82.74 | 11.67 |
| STD 10.5a0 (%) | 27.75 | 14.41 | 33.46 | 12.85 | **37.38**[B] | 13.23 | 35.93 | 13.23 |
| STD S1 (mm/s) | 114.70 | 19.89 | 109.58 | 17.82 | 108.33 | 20.39 | 108.83 | 19.51 |
| STD S2 (mm/s) | 113.15 | 11.87 | 110.63 | 29.90 | 107.30 | 32.09 | 108.50 | 31.20 |
| ACC10.0 (%) | 53.75 | 5.85 | 52.14 | 22.51 | **64.60**[B] | 20.21 | 60.22 | 21.72 |
| ACC10.5 (%) | 19.00 | 11.05 | 15.31 | 11.91 | **22.30**[B] | 13.54 | 19.88 | 13.37 |
| DA (mm) | 14.53 | 2.83 | **8.23**[A] | 3.52 | **5.63**[AB] | 2.86 | 6.62 | 3.43 |
| DA250 (mm) | 10.90 | 4.96 | **5.71**[A] | 2.78 | **4.34**[AB] | 2.34 | 4.88 | 2.68 |
| Δ S1-S2 | -1.55 | 20.63 | 1.05 | 23.75 | -1.03 | 26.98 | -0.33 | 25.82 |
| Δ HOLD10 (%) | 21.50 | 12.07 | 28.94 | 23.47 | **19.15**[B] | 18.82 | 22.52 | 20.95 |
| Δ HOLD10.5 (%) | 8.75 | 23.56 | 18.15 | 16.34 | 15.08 | 15.23 | 16.06 | 15.75 |

Notes.

[A] = differences with 8-points score

[B] = differences with 9-points score; *n* 8-score = 4; *n* 9-score = 123; *n* 10-score = 233; *n* Total number of shots = 360. *M* = Mean; *SD* = Standard deviation. Significant differences are marked in bolt letters.

differences. Lower values of STD 9a0 were observed in the 8-score than in the 9-score ($p$ = .018; $d$ = 1.36; $IC$ = 0.35–2.37) and 10-score shots ($p$ = .019; $d$ = 1.40; $IC$ = 0.4–2.39). Higher values of ACC10.0 were observed in the 10-score shots than the 9-score shots ($p$ < .001; $d$ = 0.59; $IC$ = 0.37–0.82). Higher values of ACC10.5 were observed in the 10-score shots than the 9-score shots ($p$ < .001; $d$ = 0.54; $IC$ = 0.32–0.76). Higher values of STD 10.5a0 were observed in the 10-score shots than the 9-score shots ($p$ = .02; $d$ = 0.54; $IC$ = 0.32–0.76). Higher values of DA were observed in the 8-score than the 9-score ($p$ < .001; $d$ = 1.8; $IC$ 0.78–2.82) and 10-score shots ($p$ < .001; $d$ = 3.11; $IC$ = 2.08–4.14) and in the 9-score shots than the 10-score shots ($p$ < .001; $d$ = 0.84; $IC$ = 0.61–1,07). Higher values of DA250 were observed in the 8-score than the 9-score ($p$ < .001; $d$ = 1.82; $IC$ 0.80–2.84) and 10-score shots ($p$ < .001; $d$ = 2.74; $IC$ = 1.72–3.76) and in the 9-score shots than the 10-score shots ($p$ < .001; $d$ = 0.55; $IC$ = 0.33–0.77). Higher values of Δ HOLD10 were observed in the 9-score shots than the 10-score shots ($p$ < .001; $d$ = 0.48; $IC$ = 0.26–0.7). For the rest of the comparisons, no differences were found at $p$ > .05.

The principal component analysis revealed five factors from the aiming point trajectory data ($n$ = 360), which explained 79.68% of the total variance (Table 4). The five factors were stability of hold (%), aiming accuracy, cleanness of triggering, stability of hold and aiming time.

Correlations were found between the performance and technical variables (Table 5). On the one hand, scores without decimals were related to ACC10.0 ($r$ = .27; $p$ < .001), ACC10.5 ($r$ = .24; $p$ < .001), STD 10a0 ($r$ = .13; $p$ = .017), STD 10.5a0 ($r$ = .15; $p$ = .003), DA ($r$ = -.42; $p$ < .001), DA250 ($r$ = -.31; $p$ < .001) and Δ HOLD10 ($r$ = -.21; $p$ < .001). On the other hand, the scores with decimals were related to ACC10.0 ($r$ = .28; $p$ < .001), ACC10.5 ($r$ = .28; $p$ < .001), STD 10a0 ($r$ = .15; $p$ = .005), STD 10.5a0 ($r$ = .17; $p$ = .001), DA ($r$ = -.45; $p$ < .001), DA250 ($r$ = -.34; $p$ < .001) and Δ HOLD10 ($r$ = -.21; $p$ < .001). For the significant correlations, the effect size was classified as small to medium ($r$ = .13 to .45).

**Table 4. Principal component analysis (varimax rotation) rotated solution of the aiming point variables from all the measured shots ($n = 360$).**

| | Factor 1 | Factor 2 | Factor 3 | Factor 4 | Factor 5 |
|---|---|---|---|---|---|
| | Stability of hold % | Aiming accuracy | Cleanness of triggering | Pistol's barrel speed | Aiming time |
| Eigen value | 3.1 | 1.489 | 1.348 | 1.032 | 0.999 |
| Percentage of variance | 31.001 | 14.892 | 13.477 | 10.32 | 9.988 |
| STD 10a0 (%) | 0.843 | | | | |
| STD 10.5a0 (%) | 0.833 | | | | |
| STD 9a0 (%) | 0.433 | | | | |
| ACC10.5 (%) | | 0.904 | | | |
| ACC10.0 (%) | | 0.896 | | | |
| DA250 (mm) | | | 0.894 | | |
| DA (mm) | | | 0.846 | | |
| STD S2 (mm/s) | | | | 0.874 | |
| STD S1 (mm/s) | | | | 0.834 | |
| TIME ON TARGET (s) | | | | | 0.939 |

Notes. Factor loadings of absolute value greater than 0.4 are shown.

A multiple regression analysis was performed with two criteria. According to the score without decimals criterion, the model was not significant at step 1 ($F_{(1,358)} = 1.49$; $p = .223$; $r^2 = .004$) or at step 2 ($F_{(6,353)} = 1.83$; $p = .093$; $r^2 = .03$), but it was significant at step 3 ($F_{(8,351)} = 4.05$; $p < .001$; $r^2 = .084$) and at step 4 ($F_{(10,349)} = 10.21$; $p < .001$; $r^2 = .226$). ACC10.0 was a significant predictor ($p = .023$, $\beta = 0.20$) at the third step. ACC10.0 ($p = .03$, $\beta = 0.18$), time on target ($p = .024$, $\beta = -0.11$) and DA ($p < .001$, $\beta = -0.36$) were significant predictors at the fourth step. $\Delta r^2$ was significant from step 2 to step 3 ($p < .001$) and from step 3 to step 4 ($p < .001$). According to the score with decimals criterion, the model was not significant at step 1 ($F_{(1,358)} = 0.32$; $p = .570$; $r^2 = .001$), but it was significant at step 2 ($F_{(6,353)} = 2.26$; $p = .037$; $r^2 = .021$), at step 3 ($F_{(8,351)} = 4.88$; $p < .001$; $r^2 = .100$) and at step 4 ($F_{(9,350)} = 11.66$; $p < .001$; $r^2 = .25$). DA ($p < .001$, $\beta = -0.34$) and DA250 ($p < .045$, $\beta = -0.13$) were a significant predictor at the fourth

**Table 5. Two-tailed Pearson's correlation coefficient $R$ values between shot scores with and without decimals and shooting technical variables.**

| Variable | Score without decimals | | Score with decimals | |
|---|---|---|---|---|
| | r | p | r | p |
| Time on target | -.06 | .223 | -.03 | .570 |
| STD 9a0 (%) | .04 | .415 | .06 | .275 |
| STD 10a0 (%) | **.13** | **.017** | **.15** | **.005** |
| STD 10.5a0 (%) | **.15** | **.003** | **.17** | **.001** |
| STD S1 (mm/s) | -.04 | .456 | -.03 | .520 |
| STD S2 (mm/s) | -.05 | .316 | -.06 | .242 |
| ACC10.0 (%) | **.27** | **< .001** | **.28** | **< .001** |
| ACC10.5 (%) | **.24** | **< .001** | **.28** | **< .001** |
| DA (mm) | **-.42** | **< .001** | **-.45** | **< .001** |
| DA250 (mm) | **-.31** | **< .001** | **-.34** | **< .001** |
| Δ S1-S2 | -.03 | .518 | -.05 | .354 |
| Δ HOLD10 (%) | **-.21** | **< .001** | **-.21** | **< .001** |
| Δ HOLD10.5 (%) | -.07 | .186 | -.09 | .081 |

Notes. $r$ = correlation level; $p$ = level of significance. Significant $p$ values are in bold letters.

**Table 6. Hierarchical regressions of performance variables onto the technical factors (aiming time, stability of hold, aiming accuracy and cleanness of triggering).**

| Model | Predictor | Score without decimals | | | Score with decimals | | |
|---|---|---|---|---|---|---|---|
| | | $\beta$ | $t$ | $p$ | $\beta$ | $t$ | $p$ |
| Step 1 | TIME ON TARGET (s) | -0.06 | -1.22 | 0.223 | -0.03 | -0.57 | 0,57 |
| Aiming time | F/ $R^2$/ Adj. $R^2$ | 1.49/0.004/0.001 | | 0.223 | 0.32/0.001/-0.002 | | 0.57 |
| Step 2 | TIME ON TARGET (s) | -0.06 | -1.11 | 0.267 | -0.02 | -0.47 | 0.643 |
| Stability of hold | STD 9a0 (%) | 0.02 | 0.43 | 0.667 | 0.03 | 0.63 | 0.527 |
| | STD 10a0 (%) | 0.05 | 0.63 | 0.528 | 0.07 | 0.92 | 0.361 |
| | STD 10.5a0 (%) | 0.13 | 1.75 | 0.081 | 0.14 | 1.91 | 0.057 |
| | STD S1 (mm/s) | 0.05 | 0.74 | 0.459 | 0.08 | 1.17 | 0.242 |
| | STD S2 (mm/s) | -0.04 | -0.66 | 0.512 | -0.06 | -0.92 | 0.361 |
| | F/ $R^2$/ Adj. $R^2$ | 1.83/0.03/0.014 | | 0.093 | 2.26/0.037/0.021 | | **0.037** |
| | $\Delta$ F/$\Delta$ $R^2$ | 1.89/0.026 | | 0.095 | 2.64/0.036 | | **0.023** |
| Step 3 | TIME ON TARGET (s) | -0.08 | -1.54 | 0.125 | -0.05 | -0.88 | 0.379 |
| Aiming acuracy | STD 9a0 (%) | 0.04 | 0.81 | 0.421 | 0.06 | 1.05 | 0.296 |
| | STD 10a0 (%) | 0.01 | 0.11 | 0.911 | 0.04 | 0.51 | 0.612 |
| | STD 10.5a0 (%) | 0.06 | 0.83 | 0.406 | 0.06 | 0.90 | 0.369 |
| | STD S1 (mm/s) | 0.04 | 0.57 | 0.568 | 0.07 | 1.09 | 0.275 |
| | STD S2 (mm/s) | -0.01 | -0.21 | 0.833 | -0.03 | -0.45 | 0.657 |
| | ACC10.0 (%) | 0.20 | 2.29 | **0.023** | 0.13 | 1.48 | 0.139 |
| | ACC10.5 (%) | 0.06 | 0.72 | 0.472 | 0.16 | 1.84 | 0.067 |
| | F/ $R^2$/ Adj. $R^2$ | 4.05/0.084/0.064 | | < **.001** | 4.88/0.10/0.08 | | < **.001** |
| | $\Delta$ F/$\Delta$ $R^2$ | 10.41/0.054 | | < **.001** | 12.31/0.063 | | < **.001** |
| Step 4 | TIME ON TARGET (s) | -0.11 | -2.27 | **0.024** | -0.07 | -1.57 | 0.117 |
| Cleanness of triggering | STD 9a0 (%) | 0.00 | 0.01 | 0.995 | 0.01 | 0.26 | 0.798 |
| | STD 10a0 (%) | -0.08 | -1.08 | 0.281 | -0.04 | -0.60 | 0.552 |
| | STD 10.5a0 (%) | 0.03 | 0.48 | 0.634 | 0.04 | 0.57 | 0.571 |
| | STD S1 (mm/s) | -0.02 | -0.32 | 0.753 | 0.02 | 0.28 | 0.782 |
| | STD S2 (mm/s) | 0.07 | 1.19 | 0.235 | 0.06 | 1.08 | 0.28 |
| | ACC10.0 (%) | 0.18 | 2.15 | **0.032** | 0.10 | 1.28 | 0.202 |
| | ACC10.5 (%) | 0.01 | 0.15 | 0.882 | 0.11 | 1.35 | 0.178 |
| | DA (mm) | -0.36 | -5.33 | < **.001** | -0.34 | -5.19 | < **.001** |
| | DA250 (mm) | -0.09 | -1.46 | 0.146 | -0.13 | -2.01 | **0.045** |
| | F/ $R^2$/ Adj. $R^2$ | 10.21/0.226/0.204 | | < **.001** | 11.66/0.25/0.229 | | < **.001** |
| | $\Delta$ F/$\Delta$ $R^2$ | 32.01/0.142 | | < **.001** | 34.97/0.15 | | < **.001** |

Notes. Significant $p$ values are in bold letters.

step. $\Delta r^2$ was significant from step 1 to step 2 ($p$ = .023), from step 2 to step 3 ($p$ < .001) and from step 3 to step 4 ($p$ < .001) (see Table 6).

## Discussion

The objective of the present study was to analyse what factors determine shooting performance in an elite-level air pistol shooter by using the SCATT optoelectronic device. The main results of this study showed that five components (aiming time, stability of hold, aiming accuracy, cleanness of triggering and shooting delta) determined a total of 79.68% of the shooting variance. From these factors, aiming accuracy and cleanness of triggering together explained up to 25% of the shooting performance, with cleanness of triggering being the most important factor. Additionally, significant correlations were identified between the performance and

stability of hold, aiming accuracy, cleanness of triggering and shooting delta, with effect sizes from small to medium. Lastly, significant differences were found among the 8-score, 9-score and 10-score shots, with medium to large effect sizes.

Regarding the main components of shooting variance, our results (79.68%) would be in accordance with studies that pointed out variances from 83.2% [14] to 88% [5] in air rifle shooting, from 75.8% [6] to 81.07% [7] in air pistol shooting or 78.94% in running target shooting. Interestingly, the mean variance explained by the main components of shooting were close to 81% in the previous literature, suggesting that the total performance variance in all shooting modalities was similar regardless of the number of the main technical elements. In addition, these small percentage differences between studies, which ranged from 75,8 to 88%, could be due to the participants' sport level [5], gender [24] or modality [11].

The stability of holds has been suggested to determine performance [8]. This technical factor is related to body sway [25] and could be a differential element between sport levels regarding the air rifle [5] and the air pistol [26]. Thus, elite-level shooters have better stability than novice shooters in air rifle [5] and air pistol [11]. Accordingly, stability of hold could explain in air rifle shooting up to 54% of the score variance in a sample of forty international and national-level shooters, being 18 males and 22 females [5] and specifically in air pistol shooting, a 33% for a sample of eight to ten nationally USA ranked males [6, 13], from 24% to 36% for a sample of eight national-level males and ten females) [7] or 28% for five elite-level participants distributed in two males and three females [10]. In contrast, our results showed weak relations between the stability of the hold and performance, which oscillated between 2 and 3%. Consequently, our results would be in line with those studies that revealed small performance effects of hold stability [14] or that showed no correlations [10].

These differences could be due to the weapon characteristics, the shooting position and hold points or even the effect of the specific shooting clothes on the air rifle [27]. Additionally, the elite-level shooter performance (points per shot) in previous studies 9.7 [10], 9.39 (12) or 9.7 [7] was lower than that in our study (10.08). This fact could provide another possible explanation for the result differences, suggesting diverse shooting technical strategies [10, 28] and the use of different skills in elite-level shooters [29].

Previous studies have pointed out that aiming accuracy has a great relevance in shooting performance [1]. Very strong correlations were found between aiming accuracy and performance for the air rifle [15] and air pistol [10]. Specifically, for the air rifle, between the score and the mean location of the aiming point ($r^2 = .41$) [15] and the percentage of aiming time spent inside the 10-ring during the last second ($r^2 = .64$) [5]. However, for air pistol, these correlations were usually smaller ($r^2 = .18$) [6] and ($r^2 = .47$) [7] found between the aiming time spent inside the 10 rings during the last second and for performance, respectively.

Consistent with previous studies, we found a positive relationship between ACC10.0 and performance; however, our results showed a lower correlation value ($r^2 = .08$). Differences between pistol and rifle could be associated with the weapon sights (open vs close: Open sights, combination of a bead or post in the front and a notched in the rear sight. Close sights, made of two circular lenses, globe type sight, which consists of a hollow cylinder in the front and a diopter, which is an adjustable occluder with a small hole in the rear sight), which would allow rifle shooters to have a greater range of aiming accuracy. Additionally, the rifle shooting position and the number of hold points would make it easier to stabilize the weapon within the 10-score zone. On the other hand, the differences in the pistol performance could be explained by the higher ACC10.0 percentage in our study compared with that in previous studies (60.22% vs 42% [10], 39.9% [6] or 37.6% [7]).

Although previous studies have presented contradictory results regarding the cleanness of triggering, this factor seems to be negatively related to shooting performance [1]. Cleanness of

triggering explained up to 47% of the variance in air rifle shooting [14]; however, the effect of this factor on performance would depend on the sport level, being more relevant for international (46%) than for national-level shooters (30%) [5]. Additionally, the changes in the cleanness of triggering during several seasons were also related to performance [17]. Nonetheless, cleanness of triggering would explain less variance for air pistol (23%) [7] or between 1 and 2% [6, 13] than for rifle. In this line, our results showed that the variance explained by the cleanness of triggering oscillated between 9%-20%. This percentage differences could be due to the way to measure the performance (with and without decimals) and the cleanness of triggering variable analysed, so we would be more in accordance with that of a previous study [7]. However, this huge difference between the literature suggest that future studies should be done in this topic.

On the one hand, some differences between our study and previous rifle results could be due to the trigger weight (higher in pistol than in rifle) or the target dimensions (smaller in rifle than in pistol) according to the official rules [22]. On the other hand, the differences in the pistol results could be because the NOPTEL device calculates the cleanness of triggering 200 ms before the shot while the SCATT value is 1 s to 250 ms, which could be a determinant factor. Moreover, the variables measured by both optoelectronic systems were different, for example: DA or DA250 (SCATT) vs absolute triggering value and relative triggering value (NOPTEL). Last, a critical aspect that could explain the differences could be the optoelectronic ballistic ratio. Unfortunately, this factor was not reported in previous studies.

The final technical aspects to discuss are the timing of triggering and time on target. Although Mason et al. (19) pointed out that the total sighting time was negatively related to absolute shot displacement, similar to previous studies, we did not find time on target or shooting delta to be related to performance [5–7]. Hence, elite-level shooters could use the total time on target differently from their less level sport mates [14]. Moreover, it is possible that elite shooters could reject more shots outside the optimal time range [8].

Although we analysed the preparatory season of an elite-level shooter with a large number of shots compared with other pistol studies [6, 7, 10], some limitations should be mentioned. The main limitation of this study is that it only analyses one elite-level shooter and other statistical technics like multilevel modelling could be used to improve the information given by our results. Therefore, more studies should be performed to check whether these results can be validated for elite pistol shooter collectively. Additionally, the statistical power was very limited when 8-score shots were compared with 9-score and 10-score shots due to the small number of 8-score shots ($n$ = 4 of 360 shots). As expected, elite-level pistol athletes do not usually commit this type of score shot since 8-score shots mean important technical mistakes at the elite-level. Moreover, although some extrapolation statistical techniques could be used, future studies should increase the number of participants and consequently the number of 8-score shots. Lastly, although it could be interpreted as a strength of our study, we consider that another limitation is the number of studies that use SCATT to make comparisons in elite-level pistol shooters and the F coefficient value possible effect on our results. Additionally, no studies have compared the use of the SCATT and NOPTEL simultaneously to analyse the shooting performance and to check which optoelectronic device could be better. Consequently, our results should be interpreted with caution until future studies can confirm this hypothesis.

## Conclusions

The results of the present study confirm the score variance percentage explained by the main shooting components in the previous literature [1, 6, 7, 10]. Aiming accuracy seems to be related to performance, and cleanness of triggering could be the most critical technical

element. Interestingly, stability of hold does not seem to be a determining factor of the score but could be a general prerequisite to achieving high levels of performance. Compared with previous literature, the variance explained in our results could suggest differences in the shooting technical strategies [10, 28] and skills [29] between shooter levels and modalities. Last, the results of this study could be used by coaches to compare data and elaborate specific training programmes to their air pistol shooters.

## Supporting information

**S1 Data.**
(PDF)

## Acknowledgments

We wish to thank the shooter who participate in the study.

## Author Contributions

**Conceptualization:** Daniel Mon-López, Jorge Lorenzo Calvo.

**Data curation:** Daniel Mon-López, Alfonso de la Rubia.

**Formal analysis:** Daniel Mon-López.

**Funding acquisition:** Ignacio Refoyo Roman.

**Investigation:** Daniel Mon-López, Abraham García-Aliaga.

**Methodology:** Daniel Mon-López, Alfonso de la Rubia, Jorge Acebes-Sánchez.

**Project administration:** Daniel Mon-López, Ignacio Refoyo Roman.

**Resources:** Ignacio Refoyo Roman.

**Software:** Daniel Mon-López, Abraham García-Aliaga.

**Supervision:** Daniel Mon-López, Alfonso de la Rubia, Jorge Acebes-Sánchez, Ignacio Refoyo Roman, Jorge Lorenzo Calvo.

**Validation:** Daniel Mon-López, Abraham García-Aliaga.

**Visualization:** Daniel Mon-López, Jorge Acebes-Sánchez, Jorge Lorenzo Calvo.

**Writing – original draft:** Daniel Mon-López.

**Writing – review & editing:** Daniel Mon-López, Alfonso de la Rubia, Abraham García-Aliaga, Jorge Acebes-Sánchez, Jorge Lorenzo Calvo.

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
