## [Decision Letter · Decision Letter 0]

15 Nov 2021

PONE-D-21-29416Optoelectronic analysis of technical factors and performance of top elite‐level air pistol shooting.PLOS ONE

Dear Dr. Mon-López,

Thank you for submitting your manuscript to PLOS ONE. After careful consideration, we feel that it has merit but does not fully meet PLOS ONE’s publication criteria as it currently stands. Therefore, we invite you to submit a revised version of the manuscript that addresses the points raised during the review process.

 Specifically, both reviewers considered that MINOR REVISIONS are needed in order to accept your manuscript.

We look forward to receiving your revised manuscript.

Kind regards,

Carlos Balsalobre-Fernández

Academic Editor

PLOS ONE

Journal Requirements:

Reviewers' comments:

Reviewer's Responses to Questions

**Comments to the Author**

1. Is the manuscript technically sound, and do the data support the conclusions?

Reviewer #1: Partly

Reviewer #2: Yes

2. Has the statistical analysis been performed appropriately and rigorously? 

Reviewer #1: Yes

Reviewer #2: Yes

3. Have the authors made all data underlying the findings in their manuscript fully available?

Reviewer #1: Yes

Reviewer #2: Yes

4. Is the manuscript presented in an intelligible fashion and written in standard English?

Reviewer #1: Yes

Reviewer #2: Yes

5. Review Comments to the Author

Reviewer #1: Review: Optoelectronic analysis of technical factors and performance of top elite‐level air pistol

shooting.

L. 33: “classificatory” - classification instead

L. 35: “distributed” – classified

L. 51: “simultaneously and sequentially” – can this be unpacked for the reader?

L. 100: “In this line, small relations…” – not clear.

L. 99-100: “Interestingly, there is some controversy regarding the relevance of cleanness of triggering on performance. Olsson and Laaksonen (6) showed that cleanness of triggering was negatively related to performance (r = -.48).” – this study based on the subsequent ones cited seems like an aberration rather than the rule and thus I would insert it after the studies which did find the association rather than first in line.

L. 117: “absolute shot displacement” - ?

L. 259: “Last,” should be “lastly”

L. 265-267: “suggesting that the total variance in all shooting modalities is similar regardless of the number of the main technical elements. In addition, these small differences could be due to the participants’ sport level (11), gender (23) or modality (14). – while I understand, this may require a rewrite to avoid confusing the reader.

L. 272-273: “Accordingly, stability of hold could explain up to 54% of the score variance in air rifle shooting (11) and 33% (7, 12), from 24% to 36% (6) or 28% (8) for pistol shooting” –these findings are in contrast to the current study and are attributed to expertise. Can the sample of (11, 7, 12, 6 and 8) be elaborated on?

L. 292: “weapon sights (open vs close)” – can you explain?

L. 305-306: “Nonetheless, cleanness of triggering would explain less variance for air pistol (23%) (6) or between 1 and 2% (7, 12). Hence, our results (9%-20%) would be in accordance with that of a previous study (6).” – within one modality this seems like a huge difference?

L. 317: Mason et al.

L. 326: “collectives” – collectively instead.

L. 329: Could the authors utilize an extrapolation statistical technique for the 8-score shots as currently this measurement may be unreliable.

General comment:

Possibly embed the descriptions in Table 1. Descriptions of the shooting technical variables. in the body of the text for ease of understanding when the reader is introduced to these elements.

The stats analyses are comprehensive.

Reviewer #2: The aim of the study was to capture technical factors of one elite pistol shooter using the SCATT system to investigate whether and to what extent technical factors influence pistol shooting performance. Therefore, the authors analyzed 360 shots by one elite pistol shooter (6 competitions) and showed in particular that aiming accuracy and cleanness of triggering determined shooting performance.

Title / Abstract

1) Line 25: The designation of the performance level of the pistol shooter “top elite-level shooter” sounds like a double designation to me. Elite-level or top-level is easier to understand, especially since you explicitly explain in the methods section how this term is to be classified with regard to the performance of the shooter – this is very important! Later in the methods section when describing the test subject, you only use the term elite-level. I would recommend choosing one name and use it consistently.

Introduction

1) Line 57 ff and lines 62 ff: I would rather put these two paragraphs at the end of the introduction in the paragraph from line 118 ff, since this shows weaknesses of previous studies, which you can use to reinforce your study.

2) Lines 76 ff: If possible, I would write a brief explanation of how each technical parameter (stability of hold, aiming accuracy and so on) was quantified. E.g. in line 92/93 you explained that the aiming accuracy is the percentage of aiming time spent inside the 10-ring during the last second. I would recommend doing this for every parameter.

Materials and Methods

1) Line 145: You adjusted the F coefficient to 12. Is this common in pistol shooting? Is this a high or low number?

2) Line 180: I think the statistical methods chosen are good. Have you ever thought about calculating multilevel modelling, because there you can observe individual shots within your athlete, which means that you’re not dependent on the mean value and you take into account that competitions turned out differently?

Results

3) Line 219 table 3:

a. Does table 3 display the mean values and standard deviations for all 360 shots?

b. Perhaps you can start the result section at line 201 with a sentence that table 3 shows the results of…! The text was very long and table 3 gives a better overview of your results.

4) Line 236 table 5: Maybe you can mark significant correlations so that the reader can see at a glance what became significant. You did this in table 3 and 6.

Discussion

1) Line 253: Please write in a first sentence briefly what the aim of the study was and then start with your main results.

2) Line 339 ff: Here you describe that there have been few studies that used the SCATT system. This is not a limitation of your study. For me, using the SCATT system is a strength of your study, since as I understood it right, the Noptel system is much more imprecise.

6. PLOS authors have the option to publish the peer review history of their article (what does this mean?). If published, this will include your full peer review and any attached files.

Reviewer #1: No

Reviewer #2: No

---

## [Author Response · Author response to Decision Letter 0]

26 Nov 2021

Reviewer #1: Review: Optoelectronic analysis of technical factors and performance of top elite‐level air pistol shooting.

We want to thank the reviewer commentaries to improve the quality of the manuscript.

L. 33: “classificatory” - classification instead

Response: The change has been done.

L. 35: “distributed” – classified

Response: The change has been done.

L. 51: “simultaneously and sequentially” – can this be unpacked for the reader?

Response: The change has been done.

L. 100: “In this line, small relations…” – not clear and L. 99-100: “Interestingly, there is some controversy regarding the relevance of cleanness of triggering on performance. Olsson and Laaksonen (6) showed that cleanness of triggering was negatively related to performance (r = -.48).” – this study based on the subsequent ones cited seems like an aberration rather than the rule and thus I would insert it after the studies which did find the association rather than first in line.

Response: Thank you for the commentary. According to your suggestion, we have rewritten the entire paragraph as follows: “Thus, cleanness of triggering explained 47% of the variance in the air rifle and 37% in running target shooting (14). Similarly, strong correlations were found in the air rifle, with higher values for international than national level shooters (5). Additionally, the changes in cleanness of triggering during three-seasons were related to the performance of elite-level rifle shooters (17). In contrast, Olsson and Laaksonen (7) showed that cleanness of triggering was negatively related to performance (r = -.48). In this line, weak corelations were found between the NOPTEL relative triggering value and the pistol score (6, 13).”

L. 117: “absolute shot displacement” - ?

Response: Thank you. We have added the meaning of this concept from the original research: “(sum of vertical and horizontal displacement)”.

L. 259: “Last,” should be “lastly”

Response: The change has been done.

L. 265-267: “suggesting that the total variance in all shooting modalities is similar regardless of the number of the main technical elements. In addition, these small differences could be due to the participants’ sport level (11), gender (23) or modality (14). – while I understand, this may require a rewrite.

Response: Thank you. The paragraph has been rewritten to avoid confusing the reader: “Interestingly, the mean variance explained by the main components of shooting were close to 81% in the previous literature, suggesting that the total performance variance in all shooting modalities was similar regardless of the number of the main technical elements. In addition, these small percentage differences between studies, which ranged from 75,8 to 88 %, could be due to the participants’ sport level (5), gender (24) or modality (11).”

L. 272-273: “Accordingly, stability of hold could explain up to 54% of the score variance in air rifle shooting (11) and 33% (7, 12), from 24% to 36% (6) or 28% (8) for pistol shooting” –these findings are in contrast to the current study and are attributed to expertise. Can the sample of (11, 7, 12, 6 and 8) be elaborated on?

Response: Thank you for your commentary. The sample has been detailed as follows: “Accordingly, stability of hold could explain in air rifle shooting up to 54% of the score variance in a sample of forty international and national-level shooters, being 18 males and 22 females (5) and specifically in air pistol shooting, a 33% for a sample of eight to ten nationally USA ranked males (6, 13), from 24% to 36% for a sample of eight national-level males and ten females) (7) or 28% for five elite-level participants distributed in two males and three females (10).”.

L. 292: “weapon sights (open vs close)” – can you explain?

Response: Thank you for your suggestion. We have added a explication about this technical topic: “(open vs close: Open sights, combination of a bead or post in the front and a notched in the rear sight. Close sights, made of two circular lenses, globe type sight, which consists of a hollow cylinder in the front and a diopter, which is an adjustable occluder with a small hole in the rear sight),”.

L. 305-306: “Nonetheless, cleanness of triggering would explain less variance for air pistol (23%) (6) or between 1 and 2% (7, 12). Hence, our results (9%-20%) would be in accordance with that of a previous study (6).” – within one modality this seems like a huge difference?

Response: Thank you for the commentary. We have modified the paragraph to explain this possible difference. “In this line, our results showed that the variance explained by the cleanness of triggering oscillated between 9%-20%. This percentage differences could be due to the way to measure the performance (with and without decimals) and the cleanness of triggering variable analysed, so we would be more in accordance with that of a previous study (7). However, this huge difference between the literature suggest that future studies should be done in this topic.”.

L. 317: Mason et al.

Response: The change has been done.

L. 326: “collectives” – collectively instead.

Response: The change has been done.

L. 329: Could the authors utilize an extrapolation statistical technique for the 8-score shots as currently this measurement may be unreliable.

Response: Thank you for the commentary. We agree the reviewer that the number of 8-score shots is a limitation and that some extrapolation statistical technique could be used. In addition, we think that it would be more interesting to get a greater sample than to make and extrapolation statistical technique. Moreover, our result section and the number of tables is large. For these reasons we have written this issue as a limitation and modified the following paragraph: “Additionally, the statistical power was very limited when 8-score shots were compared with 9-score and 10-score shots due to the small number of 8-score shots (n = 4 of 360 shots). As expected, elite-level pistol athletes do not usually commit this type of score shot since 8-score shots mean important technical mistakes at the elite-level. Moreover, although some extrapolation statistical techniques could be used, future studies should increase the number of participants and consequently the number of 8-score shots.”

General comment:

Possibly embed the descriptions in Table 1. Descriptions of the shooting technical variables. in the body of the text for ease of understanding when the reader is introduced to these elements.

Response: Thank you for the suggestion. We have added a brief explanation of how each technical parameter was quantified in the text to make the introduction easier for the readers. 

“One of the main shooting factors is the stability of hold or weapon stability, considered as the steadiness of the pistol barrel and measured as the mean aiming trace speed or the percentage of aiming time spent inside the 10-ring or 9-ring drawn around the hit point during the last second (5, 8).

“Interestingly, there is some controversy regarding the relevance of cleanness of triggering (movement of the aiming point during the last 0.2 s (5) or the distance between the midpoint of aim during the last interval time selected and the centre of the shot (16)) on performance.”

“The last technical aspect is related to the time to shoot, which has two components: timing of triggering (total aiming time) and time on target (aiming time spent continuously on the target).”

The stats analyses are comprehensive.

Response: Thank you for this positive commentary.

Reviewer #2: The aim of the study was to capture technical factors of one elite pistol shooter using the SCATT system to investigate whether and to what extent technical factors influence pistol shooting performance. Therefore, the authors analyzed 360 shots by one elite pistol shooter (6 competitions) and showed in particular that aiming accuracy and cleanness of triggering determined shooting performance.

Title / Abstract

1) Line 25: The designation of the performance level of the pistol shooter “top elite-level shooter” sounds like a double designation to me. Elite-level or top-level is easier to understand, especially since you explicitly explain in the methods section how this term is to be classified with regard to the performance of the shooter – this is very important! Later in the methods section when describing the test subject, you only use the term elite-level. I would recommend choosing one name and use it consistently.

Response: Thank you for the commentary. Following your suggestion, we have been consistent throughout all the manuscript using the term "elite-level".

Introduction

1) Line 57 ff and lines 62 ff: I would rather put these two paragraphs at the end of the introduction in the paragraph from line 118 ff, since this shows weaknesses of previous studies, which you can use to reinforce your study.

Response: Thank you for the commentary. We have moved and modified these two paragraphs at the end of the introduction section following your suggestion.

2) Lines 76 ff: If possible, I would write a brief explanation of how each technical parameter (stability of hold, aiming accuracy and so on) was quantified. E.g. in line 92/93 you explained that the aiming accuracy is the percentage of aiming time spent inside the 10-ring during the last second. I would recommend doing this for every parameter.

Response: Thank you for the suggestion. We have added a brief explanation of how each technical parameter was quantified. 

“One of the main shooting factors is the stability of hold or weapon stability, considered as the steadiness of the pistol barrel and measured as the mean aiming trace speed or the percentage of aiming time spent inside the 10-ring or 9-ring drawn around the hit point during the last second (5, 8).

“Interestingly, there is some controversy regarding the relevance of cleanness of triggering (movement of the aiming point during the last 0.2 s (5) or the distance between the midpoint of aim during the last interval time selected and the centre of the shot (16)) on performance.”

“The last technical aspect is related to the time to shoot, which has two components: timing of triggering (total aiming time) and time on target (aiming time spent continuously on the target).”

Materials and Methods

1) Line 145: You adjusted the F coefficient to 12. Is this common in pistol shooting? Is this a high or low number? 

Response: The F coefficient was adjusted to 12 because this is the default value give by the SCATT software. However, some coaches (included one of the researchers) usually use higher F values to improve the cleanness of the triggering in their elite-level athletes. But unfortunately, we know this data from the coaches’ experience and not from any paper. So, we decide to use the default value given by the software and most used by the shooting community. Nonetheless, taking into account your commentary we have modified the last paragraph of the discussion suggesting that this should be analysed in future studies:

“Lastly, although it could be interpreted as a strength of our study, we consider that another limitation is the number of studies that use SCATT to make comparisons in elite-level pistol shooters and the F coefficient value possible effect on our results. Additionally, no studies have compared the use of the SCATT and NOPTEL simultaneously to analyse the shooting performance and to check which optoelectronic device could be better.”

2) Line 180: I think the statistical methods chosen are good. Have you ever thought about calculating multilevel modelling, because there you can observe individual shots within your athlete, which means that you’re not dependent on the mean value and you take into account that competitions turned out differently?

Response: Thank you for the commentary. We decide to follow the analysis made by the previous literature and replicate it in the same way to be able to compare the results with exactly the same statistical analysis. As the results section were very large, we decided not to make any additional analysis. However, we agree the reviewer that this statistical analysis technique could be interesting for future researchers and for this reason we have added a sentence about this issue:

“The main limitation of this study is that it only analyses one elite-level shooter and other statistical technics like multilevel modelling could be used to improve the information given by our results. Therefore, more studies should be performed to check whether these results can be validated for elite pistol shooter collectively.”

Results

3) Line 219 table 3:

a. Does table 3 display the mean values and standard deviations for all 360 shots?

Response: Yes. We have corrected it and added the n for the total shots in the notes of the table.

b. Perhaps you can start the result section at line 201 with a sentence that table 3 shows the results of…! The text was very long and table 3 gives a better overview of your results.

Response: Thank you for the suggestion. We have started the result section whit the sentence: “Table 3 shows the comparison results of the technical variables values by shot score without decimals.”

4) Line 236 table 5: Maybe you can mark significant correlations so that the reader can see at a glance what became significant. You did this in table 3 and 6. 

Response: Thank you. We have marked the significant correlations on bold letters.

Discussion

1) Line 253: Please write in a first sentence briefly what the aim of the study was and then start with your main results.

Response: Thank you for the comment. We have added the following first sentence : “The objective of the present study was to analyse what factors determine shooting performance in an elite-level air pistol shooter by using the SCATT optoelectronic device. The main results of this study showed”

---

## [Editor Report · Decision Letter 1]

22 Dec 2021

Optoelectronic analysis of technical factors and performance of elite‐level air pistol shooting.

PONE-D-21-29416R1

Dear Dr. Mon-López,

We’re pleased to inform you that your manuscript has been judged scientifically suitable for publication and will be formally accepted for publication once it meets all outstanding technical requirements.

Kind regards,

Carlos Balsalobre-Fernández

Academic Editor

PLOS ONE
---

## [Editor Report · Acceptance letter]

28 Dec 2021

PONE-D-21-29416R1 

Optoelectronic analysis of technical factors and performance of elite‐level air pistol shooting. 

Dear Dr. Mon-López:

I'm pleased to inform you that your manuscript has been deemed suitable for publication in PLOS ONE. Congratulations! Your manuscript is now with our production department. 

Kind regards, 

on behalf of

Dr. Carlos Balsalobre-Fernández 

Academic Editor

PLOS ONE